# Metformin Protects against Diabetic Cardiomyopathy: An Association between Desmin–Sarcomere Injury and the iNOS/mTOR/TIMP-1 Fibrosis Axis

**DOI:** 10.3390/biomedicines10050984

**Published:** 2022-04-23

**Authors:** Amal F. Dawood, Norah M. Alzamil, Peter W. Hewett, Maha A. Momenah, Mohammad Dallak, Samaa S. Kamar, Dina H. Abdel Kader, Hanaa Yassin, Mohamed A. Haidara, Amro Maarouf, Bahjat Al-Ani

**Affiliations:** 1Department of Basic Medical Sciences, College of Medicine, Princess Nourah Bint Abdulrahman University, Riyadh 11671, Saudi Arabia; afdawood@pnu.edu.sa; 2Department of Physiology, Kasr al-Aini Faculty of Medicine, Cairo University, Cairo 12613, Egypt; 3Department of Clinical Science, Family Medicine, College of Medicine, Princess Nourah Bint Abdulrahman University, Riyadh 11671, Saudi Arabia; nmalzamil@pnu.edu.sa; 4Institute of Cardiovascular Sciences, College of Medicine and Dental Sciences, University of Birmingham, Birmingham B15 2TT, UK; p.w.hewett@bham.ac.uk; 5Department of Biology, College of Science, Princess Nourah Bint Abdulrahman University, Riyadh 11671, Saudi Arabia; mamomenah@pnu.edu.sa; 6Department of Physiology, College of Medicine, King Khalid University, Abha 61421, Saudi Arabia; mdallak@yahoo.com; 7Department of Medical Histology, Kasr al-Aini Faculty of Medicine, Cairo University, Cairo 12613, Egypt; dr_samaakamar@yahoo.com (S.S.K.); dinahelmy2000@yahoo.com (D.H.A.K.); 8Department of Clinical Biochemistry, Birmingham Heartlands Hospital, University Hospitals Birmingham NHS Foundation Trust, Birmingham B9 5SS, UK; a.maarouf@nhs.net

**Keywords:** diabetes, cardiomyopathy, desmin, sarcomere, metformin, fibrosis

## Abstract

The intermediate filament protein desmin is essential for maintaining the structural integrity of sarcomeres, the fundamental unit of cardiac muscle. Diabetes mellitus (DM) can cause desmin to become dysregulated, following episodes of nitrosative stress, through the activation of the iNOS/mTOR/TIMP-1 pathway, thereby stimulating collagen deposition in the myocardium. In this study, type 2 diabetes mellitus (T2DM) was induced in rats. One group of animals was pre-treated with metformin (200 mg/kg) prior to diabetes induction and subsequently kept on metformin until sacrifice at week 12. Cardiac injuries developed in the diabetic rats as demonstrated by a significant (*p* < 0.0001) inhibition of desmin immunostaining, profound sarcomere ultrastructural alterations, substantial damage to the left ventricular tissue, collagen deposition, and abnormal ECG recordings. DM also significantly induced the cardiac expression of inducible nitric oxide synthase (iNOS), mammalian target of rapamycin (mTOR), and the profibrogenic biomarker tissue inhibitor of metalloproteinase-1 (TIMP-1). The expression of all these markers was significantly inhibited by metformin. In addition, a significant (*p* < 0.0001) correlation between desmin tissue levels/sarcomere damage and glycated hemoglobin, heart rate, iNOS, mTOR, and fibrosis was observed. These findings demonstrate an association between damage of the cardiac contractile unit—desmin and sarcomere—and the iNOS/mTOR/TIMP-1/collagen axis of fibrosis in T2DM-induced cardiomyopathy, with metformin exhibiting beneficial cardiovascular pleiotropic effects.

## 1. Introduction

Type 2 diabetes mellitus (T2DM) is a chronic metabolic disorder characterized by impaired insulin secretion and/or action [1]. In recent decades, there has been an exponential rise in obesity, which has fueled an upsurge in the prevalence of diabetes mellitus (DM) resulting in a ‘diabesity’ pandemic [2]. Although estimates vary, up to 80% of individuals with DM are expected to succumb to cardiovascular disease (CVD) [3,4], which highlights the significant pressure posed by DM upon health care systems.

In addition to hyperglycemia, the majority of patients with T2DM have features of the metabolic syndrome, the components of which are independently associated with CVD [5]. These include hypertension, atherogenic dyslipidemia, abnormal fat distribution and oxidative stress [6]. Indeed, the deposition of metabolically dynamic visceral adipose tissue in T2DM is thought to play a critical role in the secretion of common pro-inflammatory adipokines such as tumor necrosis factor-alpha (TNF-α), monocyte chemoattractant protein-1, leptin, interleukin (IL-6), resistin, and plasminogen activator inhibitor-1 (PAI-1) [7,8]. Thus, central obesity, a surrogate marker of visceral adipose tissue, can be considered an important mediator of inflammation. How such pro-inflammatory mediators are triggered has not been fully elucidated. One proposed mechanism involves both oxidative as well as endoplasmic reticulum stress as critical stimulators of the inflammatory pathway [9]. High-fat diets have been shown to increase reactive oxygen species (ROS) within adipocytes via NADPH oxidase activation [10]. Indeed, dietary anti-oxidants have been shown to reverse ROS production, with resulting improvements in blood pressure and serum lipid composition [11].

The anti-diabetic drug metformin belongs to the biguanide family and has been the mainstay of T2DM treatment for the last three decades. Whilst its mechanism of action is not fully elucidated, it is thought to mainly exert its anti-hyperglycemic effect by reducing hepatic gluconeogenesis as well as by increasing peripheral tissue uptake of glucose, thus improving overall insulin resistance [12]. Metformin is also purported to reduce endothelial dysfunction, a critical process of vascular pathogenesis, thus potentially providing a therapeutic option to reduce the progression of CVD [13]. T2DM is associated with cardiac fibrosis, which may lead to impaired compliance and ventricular ejection, with eventual progression to congestive cardiac failure and dysrhythmias [14]. Although the molecular basis for cardiac fibrosis has not been fully elucidated, a cardiac insult will promote ROS activation, as observed in the hearts of diabetic rats [15], or p53-mediated apoptosis. In regard to the latter, deletion of endothelial p53 in a rat model with chronic cardiac pressure overload, prevented the progression to cardiac fibrosis and heart failure [16].

Desmin is the major protein of cardiac muscle intermediate filaments in both myocardial contractile and conducting cells [17]. The critical role of desmin in cardiac function is demonstrated by its implication in various cardiomyopathies as well as in cardiac conduction-related pathologies. Indeed, desminopathies are inherited disorders that result from mutations of the desmin (DES) genes, leading to the accumulation of aggregates of misfolded desmin [18]. Further, an association between the increase in cleaved desmin and the formation of preamyloid oligomers was reported in both human and murine models of heart failure [19].

We previously investigated the ROS–p53-collagen axis of fibrosis in a rat model with DM and left ventricular injury and showed that the anti-diabetic drug metformin can ameliorate cardiac injury [20]. However, the association between damage of the cardiac contractile unit—desmin and neighboring sarcomere—and the iNOS/mTOR/TIMP-1/collagen axis of fibrosis in rats with T2DM-induced cardiomyopathy has not been previously investigated. As a result, we investigated whether such an association can be demonstrated and whether metformin could ameliorate any of its potential deleterious effects.

## 2. Materials and Methods

### 2.1. Animals

Wistar male rats (170–200 g) were housed in a clean facility with a 12 h light/dark cycle. Animals had free access to water and food. The research ethical committee at Princess Nourah Bint Abdulrahman University approved all the protocols (protocol number H-01-R-059) in accordance with the Guide for the Care and Use of Laboratory Animals published by the US National Institutes of Health (NIH publication No. 85–23, revised 1996).

### 2.2. Experimental Design

Animal work was performed as previously described [20]. Briefly, 24 rats were allocated into three groups. These consisted of non-diabetic rats (control group) and rats with T2DM but not treated with metformin (T2DM group). DM was induced using a high-carbohydrate and -fat diet (HCFD) and streptozotocin injection as previously described [21]. These rats were fed continuously on an HCFD until the end of week 12. The third group consisted of T2DM rats that were treated with metformin (Met + T2DM group). This group were given metformin (200 mg/kg) from day 1 and were fed the HCFD for two weeks before inducing T2DM. The rats continued receiving metformin and were fed the HCFD for another 10 weeks. At the end of the experiment, blood samples were collected under anesthesia, before the rats were culled by cervical dislocation. Cardiac tissue samples were then collected and analyzed. DM was confirmed in the model group one week post diabetic injection by measuring fasting blood glucose levels (>200 mg/dL) using a Randox reagent kit (Randox Laboratories Ltd., Crumlin, UK). 

### 2.3. Determination of Blood Levels of Glucose and HbA1c

The serum levels of glucose were determined colorimetrically using a Randox reagent kit (Randox Laboratories Ltd., Crumlin, UK). The blood levels of HbA1c were measured using the ELISA kit Cat. # 80300; Crystal Chem, Inc., Elk Grove Village, IL, USA).

### 2.4. Histological Examination

Left ventricle specimens were fixed in 10% formalin for 24 h prior to dehydration with an ascending grade of alcohols followed by clearing and embedding of the samples in paraffin, with the use of standard methods. Paraffin blocks were sectioned in samples of 5 μm thickness, with sections being de-paraffinized, rehydrated, and subjected to hematoxylin and eosin (H&E) staining in order to elucidate the cardiac architecture and structural changes. Cardiac sections were also stained with Masson’s trichrome, which stains collagen in blue, to demonstrate collagen deposition as a marker of cardiac fibrosis.

### 2.5. Immunohistochemistry of iNOS and Desmin

Immunohistochemical staining of left ventricle specimens was performed as we recently reported [20]. Antigen retrieval was carried out by boiling the sections in 10 mM citrate buffer (pH 6), followed by incubation with anti-iNOS antibodies (Abcam, cat # ab15323, Cambridge, UK) and anti-desmin antibodies (Cat # ab131442, Abcam, Cambridge, UK). The sections were incubated with secondary antibodies for 30 min at room temperature. The sections were counter stained with Meyer’s hematoxylin.

Morphometry of the areas% of collagen deposition and the areas% of iNOS and desmin immunostaining was performed using a “Leica Qwin 500 C” image analyzer (Cambridge, UK) in 10 non-overlapping fields for each group. Quantitative data were summarized as means and standard deviations (SD) and compared using analysis of variance (ANOVA) followed by post-hoc analysis (Tukey test). A *p*-value < 0.05 was considered statistically significant. Calculations were carried out using a statistical package of social science (SPSS) software, version 19.

### 2.6. Transmission Electron Microscopy (TEM)

Small pieces (~1 mm^3^) of the left ventricle were fixed at 4 °C in 4% buffered glutaraldehyde (SERVA, Frankfurt, Germany) with 0.2 M cacodylate buffers (TAAB essential for microscopy, Aldermaston, Berks, UK) and processed for TEM as we previously described [22]. Contrasted ultrathin sections with uranyl acetate and lead citrate (Loba Chemie Pvt. Ltd., Colaba, Mumbai, India) were examined and photographed using a Philips EM 208S TEM (FEI Company, Eindhoven, The Netherlands) at an accelerating voltage of 80 kV.

### 2.7. Western Blotting Analysis of mTOR and AMPK

Proteins were extracted from cardiac tissues, and 20 μg of protein per sample was immunoblotted as previously described [23]. The membranes were probed with anti-mTOR-phospho-Ser2448, anti-AMPK-phospho-Thr172, and beta actin antibodies (Cell Signaling Technology, Danvers, MA, USA) at 4 °C overnight. Proteins were visualized using the ECL detection kit (Merck Life Science, Gillingham, Dorset, UK). Relative expression was determined using an image analysis software to determine the band intensity of the target proteins with respect to a control sample after normalization by β-actin on the Chemi Doc MP imager.

### 2.8. Cardiac TIMP-1 Real-Time Polymerase Chain Reaction (qPCR)

qPCR was performed using primers specific for TIMP-1(sense, 5′-GGT TCC CTG GCA TAA TCT GA-3′; antisense, 5′-GTC ATC GAG ACC CCA AGG TA-3′) or β-actin as previously described [23].

### 2.9. Heart Rate (HR) and Electrocardiograph (ECG) Recordings

Heart rate (HR) and blood pressure (BP) were measured using a BP monitor (LE 5001, LETICIA scientific Instruments, Barcelona, Spain) from the tail of conscious rats applying the tail-cuff technique. The animals were warmed for 30 min at 28 °C in a thermostatically controlled heating cabinet (Ugo Basile, Gemonio, VA, Italy) for better detection of the tail artery pulse. The tail was passed through a miniaturized cuff and a tail-cuff sensor that was connected to an amplifier (LE 5001, LETICIA scientific Instruments, Barcelona, Spain). The cuff was attached to a tail cuff sphygmomanometer, and blood pressure and heart rate were recorded on a chart, with the mean of at least three measurements obtained on each occasion. ECG recordings were performed on anesthetized rats using Power lab T 26, ML 856 (AD Instruments, Bella Vista, New South Wales, Australia).

### 2.10. Statistical Analysis

The data are expressed as the mean ± SD. Data were processed and analyzed using SPSS version 10.0 (SPSS, Inc., Chicago, IL, USA). One-way ANOVA was performed followed by Tukey’s post hoc test. Pearson correlation statistical analysis was carried out for the detection of a probable significance between two different parameters. The results were considered significant if *p* ≤ 0.05.

## 3. Results

### 3.1. The Induction of Diabetic Cardiomyopathy Is Inhibited by Metformin

Diabetes is a recognized risk factor for the development of cardiomyopathy [24]. To determine whether metformin is able to prevent the development of morphological changes that lead to cardiomyopathy, we used a rat model of T2DM [20]. To investigate the structural changes in the myocardium, we assessed the integrity of cardiomyocytes using basic histological staining and TEM analyses. Representative H&E-stained left ventricular sections of the diabetic group (Figure 1B) revealed disorganized dilated cardiac myocytes (wavy arrows) displaying darkly stained nuclei (D) and the presence of spaces between cardiac cells (S) compared with the control group (Figure 1A). Treatment of the diabetic rats with metformin (Figure 1C,D) provided substantial protection against diabetes-induced cardiomyopathy, as demonstrated by a decrease in cardiomyocyte enlargement and of the spaces between the cardiac cells. As expected, metformin treatment also significantly (*p* < 0.01) decreased glycemia (Figure 1E,F) and dyslipidemia (data not shown).

Representative TEM images of the left ventricular sections obtained from the control group of rats (Figure 2A,B) demonstrated a normal cardiomyocyte ultrastructure, as shown by an oval nucleus (N) with a clear nucleolus (nu), dispersed chromatin (chr), and an intact nuclear envelope (ne). Parallel myofibrils formed regular successive sarcomeres delineated by dark Z discs (Z) with the H band (H) in the middle of the sarcomeres. Longitudinally arranged interfibrillar mitochondria (m) were densely packed between the myofibrils. The cardiac dyad tubular system was visible at the level of the Z lines. In contrast, representative TEM images of the left ventricular sections of T2DM rats (Figure 2C,D,G) showed cardiomyocytes with irregular nuclei (N), clumped chromatin (chr), and an indented nuclear envelope (ne). Discontinuous irregularly arranged myofibrils with unusually thickened Z discs, and numerous irregular bizarrely shaped mitochondria (m1), some with disrupted cristae were also observed. Note the widely separated festooned sarcolemma (arrows).

Metformin treatment prevented the development of these characteristic morphological changes in diabetic rats (Figure 2E–G). Left ventricular sections from these animals showed normal features, i.e., oval nuclei (N) with a clear nucleolus (nu) and an intact nuclear envelope (ne). However, some peripherally arranged chromatin clumps (chr) were present. Parallel widely separated myofibrils formed regular sarcomeres demarcated by dark Z discs (Z) and a clear H Band (H). Few vacuolations (V) and intact interfibrillar mitochondria (m) were seen between the myofibrils.

### 3.2. Downregulation of Cardiac Desmin Expression in Diabetic Rats Is Inhibited by Metformin

Cardiomyopathies associated with desmin abnormalities are well known [25]. In light of our findings and of the established physical connection between desmin and sarcomeres [26], we assessed the expression of desmin protein in all animal groups. Immunohistochemical staining of left ventricular tissue samples prepared from the untreated diabetic group (T2DM) 10 weeks post diabetic induction showed a substantial decrease in the expression of desmin protein (Figure 3B,D), which was significantly (*p* < 0.0001) prevented by metformin treatment (Figure 3C,D) that led to levels comparable to those of the control group (Figure 3A,D).

We further monitored the functional changes in the myocardium caused by diabetic cardiomyopathy. Electrocardiograms (ECGs) of diabetic rats revealed an abnormal ECG pattern and a decreased heart rate (Figure 3F,H). These changes were prevented in the metformin-treated group (Figure 3G,H), which produced ECG traces comparable with those of the control group (Figure 3E,H). A significant (*p* < 0.0001) correlation was obtained between desmin and heart rate (Figure 3I), as well as between desmin and sarcomere damage (Figure 3J).

### 3.3. Induction of Cardiac Fibrosis by Diabetes Is Inhibited by Metformin

Associations have been made between desmin, sarcomere damage, and fibrosis [27,28]. Therefore, taking into account our results described above showing inhibition of desmin expression and disorganization of cardiomyocyte sarcomere secondary to DM, we assessed cardiac fibrosis levels in the presence and absence of metformin in relation to glycemia, desmin levels, sarcomere damage, and cardiac function (Figure 4). Compared with the minimal collagen staining in sections prepared from the left ventricle of the control group (Figure 4A), Masson’s trichrome-stained left ventricular sections of the untreated diabetic group revealed substantial coarse collagen deposition in the perivascular spaces as well as between myocytes (Figure 4B,D). This was significantly (*p* < 0.0001) inhibited by metformin (Figure 4C,D). A significant (*p* < 0.0001) positive correlation was observed between collagen deposition and glycemia (Figure 4E) and sarcomere injury (Figure 4G), whereas a significant (*p* < 0.0001) negative correlation was observed between collagen deposition and both desmin levels (Figure 4F) and heart rate (Figure 4H).

### 3.4. Diabetes Induces the iNOS–mTOR–TIMP-1 Axis of Fibrosis Is Protected by Metformin

In order to identify an association between the iNOS–mTOR–TIMP-1 axis of fibrosis and the levels of the intermediate filament (desmin) that regulates the contractile unit within cardiomyocytes, we assessed the levels of iNOS, mTOR, TIMP-1, and the tissue protective enzyme AMP-activated protein kinase (AMPK), which is well-known to be activated by metformin [29], in the cardiac tissue of all animal groups. While negative or very weakly iNOS-immunostained cells were found in the control group (Figure 5A), immunohistochemical staining for iNOS in cardiac sections of the untreated diabetic group (T2DM) showed numerous iNOS-positive cells (Figure 5B,D), whose presence was significantly (*p* < 0.0001) reduced by metformin in the Met + T2DM group (Figure 5C,D).

The levels of phosphorylated AMPK at T172 (p-AMPK) and mTOR at S2448 (p-mTOR) as well as of TIMP-1 mRNA were evaluated in the rats by Western blotting and qPCR, respectively. In the T2DM group, there was a significant inhibition of AMPK (Figure 5E,F) and an augmentation of mTOR (Figure 5E,G) activity and TIMP-1 expression (Figure 5H) compared with the control animals. These DM-associated changes were significantly (*p* < 0.0001), but not completely, prevented by metformin treatment. We further determined the correlations between the desmin and sarcomere scores and the cardiac levels of iNOS, AMPK, mTOR, and TIMP-1 (Figure 5I–L). This was important to draw a correlation between these biomarkers and the pathogenesis of DM-induced desmin and sarcomere damage and cardiomyopathy. The desmin scores displayed a negative correlation with the levels of iNOS (r = −0.711; *p* < 0.001, Figure 5I), p-mTOR (r = −0.922; *p* < 0.0001, graph not shown), and TIMP-1 (r = −0.869; *p* < 0.0001, graph not shown), whereas p-AMPK level was positively correlated with the desmin score (r = 0.937; *p* < 0.0001, Figure 5J). Sarcomere injury scores displayed a positive correlation with mTOR (r = 0.937; *p* < 0.0001, Figure 5K) and TIMP-1 (r = 0.876; *p* < 0.0001, Figure 5L) levels.

## 4. Discussion

The present findings demonstrate that in a rat model of T2DM-induced cardiomyopathy, there is evidence of an association between damage of the cardiac contractile unit—sarcomere and desmin—and the iNOS/mTOR/TIMP-1/collagen axis of fibrosis. In addition, we showed that the anti-diabetic drug metformin is able to substantially protect against the upregulation of the investigated axis of fibrosis as well as against injuries to the cardiac contractile unit and cardiomyocytes caused by DM (Figure 6). This further corroborates our recent report on the protective effect of metformin on diabetic cardiac injury via a different cell signaling axis [20].

The major intermediate filament protein desmin connects the cardiac contractile unit sarcomere with the mitochondria, nucleus, and postsynaptic membrane of the motor end plate in order to maintain structural integrity and function of the cardiac muscle cell [26]. In addition, desmin is essential in maintaining elastic strength and integrity of sarcomeres (myofibrils) [30]. A mouse model of a desmin mutation showed sarcomere misalignment at the Z-line level [31]. However, little is known about the impact of T2DM on the integrity of the contractile unit. Interestingly, induction of type 1 diabetes mellitus (T1DM) in rats was reported to cause a disordered and irregular pattern of desmin arrangement in cardiomyocytes one year post DM induction [32]. In addition, the time interval to reach (i) the instant of maximal shortening velocity, (ii) the peak of contraction, (iii) the instant of maximal relaxation velocity, and (iv) the 50% resting left ventricle sarcomere length were also prolonged in T1DM rats compared with the control group [33]. Further, ultrastructural damage to the sarcomere rat rectus abdominis muscle has been reported in T1DM pregnant rats [34]. These reports corroborate our own study findings of T2DM-induced pathological insults such as alterations to the cardiac sarcomere ultrastructure and inhibition of desmin expression (Figure 2 and Figure 3). Sarcomere dysfunction due to metabolic disturbances has also been reported in a transgenic mouse model of lipotoxic diabetic cardiomyopathy caused by an increase in myocardial free fatty acid uptake independent of intracellular calcium levels [35]. This study also corroborates our own finding of an association between dyslipidemia (data not shown) and cardiac sarcomere damage (Figure 2).

iNOS is colocalized with desmin after human heart failure [36] and inhibition of elevated mTORC1 expression with rapamycin decreasing the number of myocytes with abnormal desmin [37]. In addition, desmin deletion and sarcomere mutations are reported to induce skeletal and cardiac muscle fibrosis [27,28]. These findings appear consistent with our data, which similarly demonstrate such events in a rat model of T2DM-induced cardiomyopathy. In addition, our data suggest that amelioration of cardiac fibrosis with metformin—via the inhibition of iNOS/mTOR/TIMP-1 cell signaling—mirrors the results of other studies in which metformin inhibited cardiac fibrosis, the latter being induced by either pressure overload [38] or T2DM [20] via the inhibition of TGFβ1/Smad3 and ROS/p53 cell signaling, respectively.

In addition to our own findings of attenuation of cardiac fibrosis, metformin is also thought to have pleiotropic effects by virtue of its anti-inflammatory properties [39]. For example, studies have demonstrated that metformin attenuates oxidized lo- density lipoprotein (LDL) or lipopolysaccharide (LPS)—induced pro-inflammatory responses in macrophages and monocytes—thus contributing to anti-atherosclerotic effects [40]. Furthermore, there is some evidence suggesting that an increased macrophage burden within cancerous tissue can lead to progression of the disease [41]. Since metformin is also thought to target macrophage differentiation, its utility as an anti-neoplastic drug is also possible, although robust clinical trials are warranted. In healthy individuals, therefore, it may be possible to exploit metformin’s pleiotropic benefits to ameliorate some of the common ageing-related comorbidities, such as obesity, MetS, diabetes, CVD, cognitive decline, and malignancies [42].

In recent years there has been great interest in the extra-antihyperglycemic effects of novel drugs, such as incretin-based therapies and sodium-glucose cotransporter-2 (SGLT-2) inhibitors. These have shown great promise in human cardiovascular outcome trials and may likely prove to be superior to metformin [43,44,45,46]. Using a rat model to determine the likely signaling pathways for these novel agents is essential to further our understanding of the underlying mechanisms as well as provide an opportunity for targeted drug therapy.

In conclusion, our study demonstrated that T2DM deleteriously impacts upon the cardiac contractile unit, upregulating iNOS/mTOR/TIMP-1-mediated fibrosis. Importantly, this effect was inhibited by metformin for a duration of 12 weeks in a T2DM rat model.

### Limitations of the Study

In addition to evaluating basic physiological parameters (pulse and ECG trace), a transthoracic echocardiogram would have been invaluable in the assessment of cardiac structure and functionality, providing more strength to the observed findings.

## Figures and Tables

**Figure 1 biomedicines-10-00984-f001:**
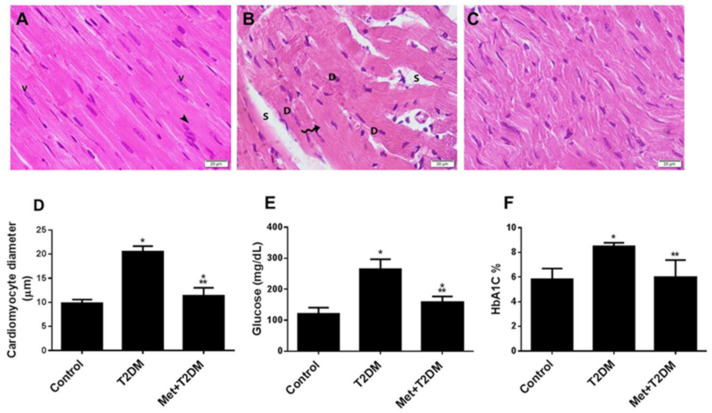
Metformin (Met) protects the cardiac architecture against injuries secondary to T2DM. H&E-stained images (400×) of left ventricles obtained 10 weeks post diabetic induction from the control untreated (**A**), T2DM (**B**), and Met + T2DM (**C**) groups are illustrated (Scale = 20 μm). Note that the arrowhead in (**A**) points to the intercalated disc, and the wavy arrow in (**B**) points to dilated cardiac myocytes. V: vesicular nuclei; S: spaces between cardiac cells; D: darkly stained nuclei; H&E: hematoxylin and eosin; T2DM: type 2 diabetes mellitus. Histograms in (**D**) represent a quantitative analysis of cardiomyocytes injury assessed on the basis of the cardiomyocyte diameter in μm in the groups described above. (**E**,**F**) Blood levels of glucose (**E**) and glycated hemoglobin (**F**) were measured at the end of the experiment, at week 12. The presented *p* values are all significant. * *p* < 0.05 versus control, ** *p* < 0.01 versus T2DM.

**Figure 2 biomedicines-10-00984-f002:**
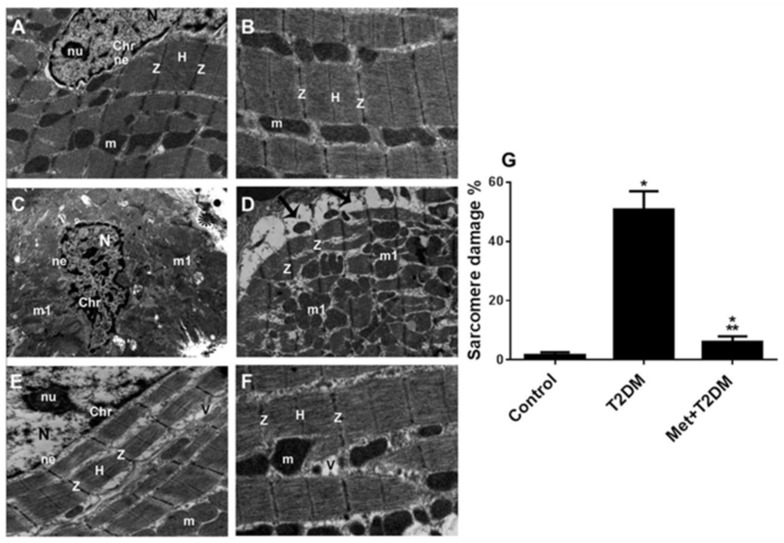
Metformin (Met) protects the cardiac ultrastructure against alterations secondary to T2DM. TEM images (10,000×) of left ventricles obtained 10 weeks post diabetic induction of untreated control (**A**,**B**), T2DM (**C**,**D**), and Met + T2DM (**E**,**F**) groups are illustrated. Note that the asterisk in (**C**) points to degenerated fragmented myofibrils, and the arrows in (**D**) point to the widely separated festooned sarcolemma. N: nucleus; nu: nucleolus; chr: chromatin; ne: nuclear envelope; V: vacuoles; m: mitochondria; m1: damaged mitochondria; H: H band; Z: Z discs; T2DM: type 2 diabetes mellitus. The histogram in (**G**) represents a quantitative analysis of % sarcomere damage in cardiac sections from the groups above. Presented *p* values are all significant. * *p* < 0.05 versus control, ** *p* < 0.0001 versus T2DM.

**Figure 3 biomedicines-10-00984-f003:**
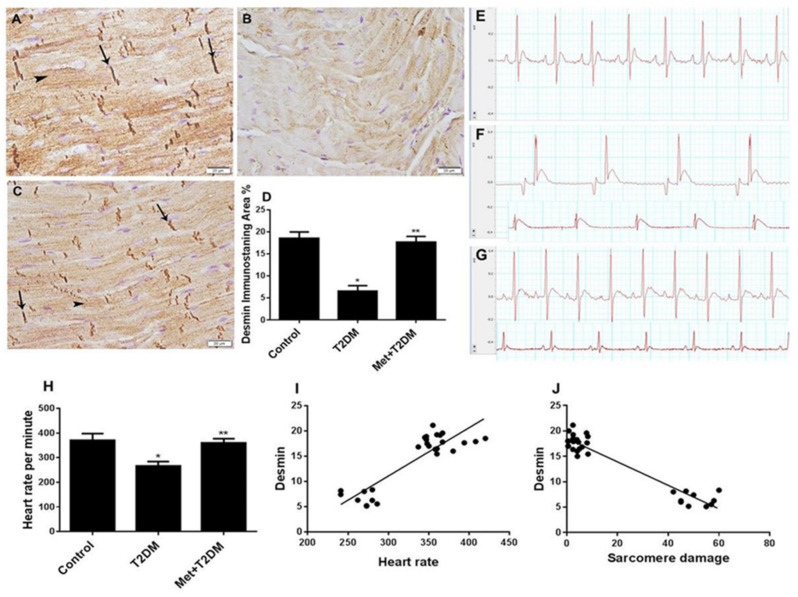
Metformin (Met) protects against the inhibition of cardiac desmin expression and ECG abnormalities caused by T2DM. Immunohistochemistry of desmin (400×) in left ventricle sections obtained 10 weeks post diabetic induction from the untreated control (**A**), T2DM (**B**), and Met + T2DM (**C**) groups are depicted (Scale = 20 μm). Note that the arrows in (**A**,**C**) point to desmin immunostaining in the intercalated discs, and the arrowheads in (**A**,**C**) point to the organized desmin distribution in the cardiomyocytes. Histograms in (**D**) represent a quantitative analysis of desmin-immunostained area % in cardiac sections from the above groups. Representative ECG recordings from the untreated control (**E**), T2DM (**F**), and Met + T2DM (**G**) groups are depicted. The heart rate of the groups was recorded at the end of the experiment (**H**). (**I**,**J**) Correlation between desmin and heart rate (**I**) and sarcomere damage (**J**). T2DM: type 2 diabetes mellitus; ECG: electrocardiographs. Presented *p* values are all significant. * *p* < 0.0001 versus control, ** *p* < 0.0001 versus T2DM.

**Figure 4 biomedicines-10-00984-f004:**
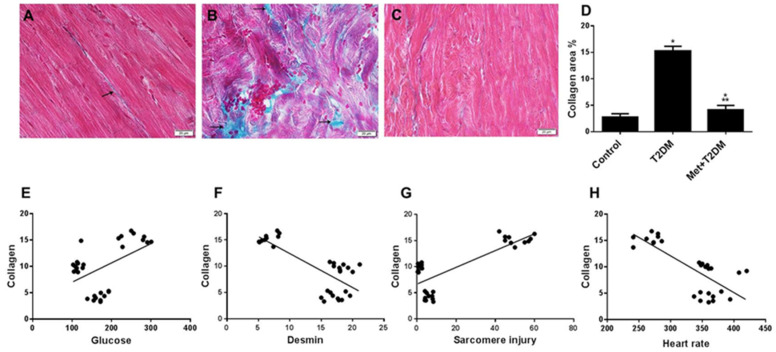
Metformin (Met) protects against cardiac fibrosis secondary to T2DM. Masson’s trichrome-stained images (400×) of left ventricles obtained 10 weeks post diabetic induction from the untreated control (**A**), T2DM (**B**), and Met + T2DM (**C**) groups are visualized using light microscopy (Scale = 20 μm). Note that the arrows point to collagen deposition. Histograms in (**D**) represent a quantitative analysis of collagen area % in cardiac sections from the above groups. (**E**–**H**) Correlation between collagen deposition (fibrosis) score and glucose levels (**E**), desmin (**F**), sarcomere injury (**G**), and heart rate (**H**). The presented *p* values are all significant. * *p* < 0.01 versus control, ** *p* < 0.0001 versus T2DM. T2DM: type 2 diabetes mellitus.

**Figure 5 biomedicines-10-00984-f005:**
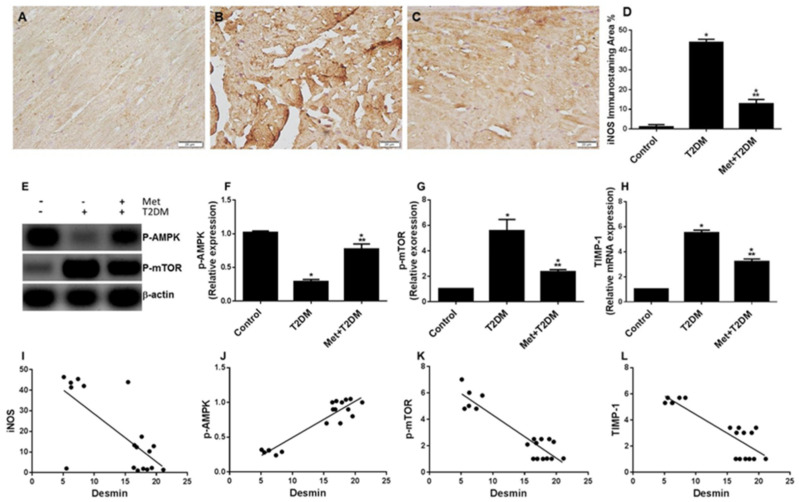
Metformin (Met) protects against the inhibition of cardiac desmin expression and ECG abnormalities caused by T2DM. iNOS immunohistochemistry of left ventricle sections (400×) obtained 10 weeks post diabetic induction from the untreated control (**A**), T2DM (**B**), and Met + T2DM (**C**) groups are depicted (Scale = 20 μm). Histograms in (**D**) represent a quantitative analysis of desmin immunostaining area % in cardiac sections from the above groups. Cardiac lysates prepared from the above groups were immunoblotted with antibodies against p-AMPK, p-mTOR, and β-actin as a loading control (**E**). The relative expression of these signaling proteins is shown (**F**,**G**). Histograms in (**H**) display the relative mRNA expression of TIMP-1 in all animal groups. In **E**, --, represent the Control group; -+, represent T2DM group; and ++, represent the treated group. The presented *p* values are all significant. * *p* < 0.01 versus control, ** *p* < 0.0001 versus T2DM. (**I**–**L**) Correlation between either the desmin score versus iNOS (I) and p-AMP (**J**) or the score of sarcomere damage versus p-mTOR (**K**) and TIMP-1 (**L**). iNOS: inducible nitric oxide synthase; T2DM: type 2 diabetes mellitus; AMPK: AMP-activated protein kinase; mTOR: mammalian target of rapamycin; TIMP-1: tissue inhibitor of metalloproteinase-1.

**Figure 6 biomedicines-10-00984-f006:**
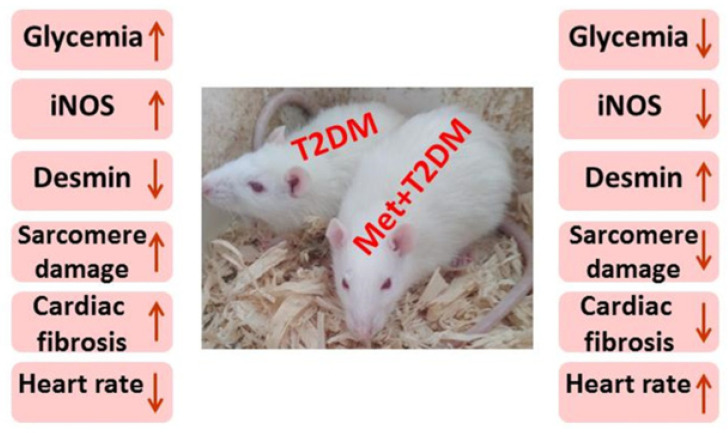
The proposed model for diabetes-induced cardiomyopathy (left side) appears inhibited by metformin (right side). ↑ = increased; ↓ = decresed. iNOS: inducible nitric oxide synthase; T2DM: type 2 diabetes mellitus; Met: metformin.

## Data Availability

The data that support the findings of this study are available on request from the corresponding author.

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
