# Peer review of "Metformin Protects against Diabetic Cardiomyopathy: An Association between Desmin–Sarcomere Injury and the iNOS/mTOR/TIMP-1 Fibrosis Axis"

_biomedicines, 2022, doi:10.3390/biomedicines10050984_

Round 1

Reviewer 1 Report

I enjoyed a review this manuscript. The paper is interesting, quite original and well written. The conclusions are supported by the results. Figures are clear.

This reviewer raises only a few issues to address.

1- Metformin in recent years is at the center of scientific research and is experiencing a second youth. The numerous extra-antihyperglycemic ancillary effects of metformin should be at least briefly discussed in both the introduction and discussion. In particular, the protective mechanisms on the endothelium, as anti-aging (Diabetes Res Clin Pract. 2020 Feb;160:108025. doi: 10.1016/j.diabres.2020.108025.), and as an adjuvant to antineoplastic chemotherapy should be addressed by the authors. 

2- At the end of the discussion the paragraph on the limitations of the study is missing. Authors should add it

Reviewer 2 Report

Dear authors, your manuscript deals on effects of Metformin treatment on  damage of the cardiac contractile unit, namely desmin and sarcomere, and iNOS/mTOR/TIMP-1/collagen axis of fibrosis in T2DM-induced cardiomyo-
pathy, You were able to demonstrate Metformin's well known pleiotropic effects on the cardiovascular  system.

Results are reliable and presented in a well organized order. The combination of WD feeding and streptozotozin injection is mixing up different diabetes pathologies (Insulinresistance and ß-cell loss) but the rats still remain metformin responsive and thus this co-treatment is an elegant way to shorten up the feeding period. Several other working groups have implementetd a combined tretament.

To the reviewer's point of view few things have to be clarified to optimize the story:

1) Justify the metformin dose In your previous paper you used 250 mg/kg BW which is even higher than the 200 mg used here. Compared to human dosing this is rather high. Are there any measurements of urine clearance, BZ profiles etc available

2) minor comments:

greek symbols are not correctly displayed in the pdf (e.g. line 142, 151, 357), use "'" instead of "´" in oligonucleotide description, in some lines the seems to be a coloration, marks (e.g. 177-178)
